# SPR and Double Resonance LPG Biosensors for *Helicobacter pylori* BabA Antigen Detection

**DOI:** 10.3390/s24072118

**Published:** 2024-03-26

**Authors:** Georgi Dyankov, Tinko Eftimov, Evdokiya Hikova, Hristo Najdenski, Vesselin Kussovski, Petia Genova-Kalou, Vihar Mankov, Hristo Kisov, Petar Veselinov, Sanaz Shoar Ghaffari, Mila Kovacheva-Slavova, Borislav Vladimirov, Nikola Malinowski

**Affiliations:** 1Institute of Optical Materials and Technologies “Acad. J. Malinowski” (IOMT), Bulgarian Academy of Sciences (BAS), 109 “Acad. G. Bonchev” Str., 1113 Sofia, Bulgaria; gdyankov@iomt.bas.bg (G.D.); vhamankov@gmail.com (V.M.); hristokisov@iomt.bas.bg (H.K.); pepi99pepi@gmail.com (P.V.); malinowski.nikola@gmail.com (N.M.); 2Central Laboratory of Applied Physics, Bulgarian Academy of Sciences, 61 Sankt Petersburg Blvd., 4000 Plovdiv, Bulgaria; tinko.eftimov@uqo.ca; 3Photonics Research Center, Université du Québec en Outaouais, Rue 101 St-Jean Bosco, Gatineau, QC J8X 3G5, Canada; shoargha@ualberta.ca; 4The Stephan Angeloff Institute of Microbiology, Bulgarian Academy of Sciences, 26 Acad. G. Bonchev Str., 1113 Sofia, Bulgaria; hnajdenski@gmail.com (H.N.); vkussovski@gmail.com (V.K.); 5National Center of Infectious and Parasitic Diseases, 44A “Gen. Stoletov” Blvd., 1233 Sofia, Bulgaria; petia.d.genova@abv.bg; 6Department of Electrical and Computer Engineering, University of Alberta, 116 St & 85 Ave, Edmonton, AB T6G 2R3, Canada; 7Department of Gastroenterology, University Hospital Tsaritsa Ioanna-ISUL, Medical University Sofia, 8 “Byalo More” Str., 1527 Sofia, Bulgaria; kovacheva_mila@abv.bg (M.K.-S.); vladimirov.borislav@gmail.com (B.V.)

**Keywords:** *Helicobacter pylori* diagnosis, outer membrane protein BabA, blood antigen, surface plasmon resonance (SPR), double resonance long period grating (DR LPG), biosensors

## Abstract

Given the medical and social significance of *Helicobacter pylori* infection, timely and reliable diagnosis of the disease is required. The traditional invasive and non-invasive conventional diagnostic techniques have several limitations. Recently, opportunities for new diagnostic methods have appeared based on the recent advance in the study of *H. pylori* outer membrane proteins and their identified receptors. In the present study we assess the way in which outer membrane protein–cell receptor reactions are applicable in establishing a reliable diagnosis. Herein, as well as in other previous studies of ours, we explore the reliability of the binding reaction between the best characterized *H. pylori* adhesin BabA and its receptor, the blood antigen Le^b^. For the purpose we developed surface plasmon resonance (SPR) and double resonance long period grating (DR LPG) biosensors based on the BabA–Le^b^ binding reaction for diagnosing *H. pylori* infection. In SPR detection, the sensitivity was estimated at 3000 CFU/mL—a much higher sensitivity than that of the RUT test. The DR LPG biosensor proved to be superior in terms of accuracy and sensitivity—concentrations as low as 10^2^ CFU/mL were detected.

## 1. Introduction

According to the World Health Organization *H. pylori* colonizes the gastric epithelium of about 50% of the world’s population contributing to approximately 75% of the stomach cancer cases and 5.5% of all types of cancer. Therefore, timely and reliable diagnosis of the infection is highly desirable. Conventional diagnostic methods have an important role in medical practice, but possess shortcomings that limit their application.

The ability of *H. pylori* to colonize gastric epithelial cells and persist in spite of the immune mechanisms targeting to eradicate it are due to its large array of outer membrane proteins. Their mechanisms of biochemical interactions are well studied and new diagnostic methods have been proposed on this basis. In this paper we explore to what extent the reaction between the *H. pylori* adhesin BabA and its receptor, the blood antigen Le^b^, can be used for diagnosing *H. pylori* infection by means of biosensors.

### 1.1. Current Diagnostic Methods

The urea breath test (UBT) is often considered as the gold standard test in the diagnosis of *H. pylori* infection. However, it is a well-known fact that the stomach is colonized by many other urease producing bacteria, which makes UBT questionable. Also, proton pump inhibitors and drugs treating *H. pylori* infection may produce false-negative results.

The significance of serology in *H. pylori* detection has been discussed for decades. Antibodies to *H. pylori* appear in the blood 3–4 weeks following infection. They can be found for a prolonged period (up to five years after infection) and after the total eradication of *H. pylori*. However, serology sensitivity and specificity is considered to be insufficient: meta-analysis has shown average sensitivity of about 80% and specificity around 85% [1].

For all these reasons, the Maastricht II Consensus Report in 2000 rejected serology as a recommended method for initial diagnosis of *H. pylori* infection in the absence of endoscopy [2]; the Maastricht III Consensus Report in 2005 suggested that “some serological tests with good sensitivity and specificity can be performed in order to establish the initial diagnosis of infection with *H. pylori*” [3]; Maastricht IV/Florence Consensus Report stated that only validated commercial tests should be used [4]; the Maastricht VI/Florence Consensus Report drew the following conclusions: (i) functional serology has shown high level of accuracy (96%) and very high negative predictive value; (ii) serology cannot be used for testing the eradication success; (iii) serology does not indicate an active infection [5]. A pragmatic approach involves utilizing serological detection of *H. pylori*, while any positive test results have to be confirmed by another, more reliable test.

The stool antigen test (SAT) is commonly used as a screening non-invasive test for *H. pylori* detection and is suitable after eradication. SAT is assessed by enzyme immunoassay (monoclonal or polyclonal) or immunochromatography. Antibiotics, proton pump inhibitors and bismuth can decrease the bacterial load and lead to false negative results [5].

Histology is one of the most commonly used diagnostic methods, although certain limitations are inherent to it. False negative results in histological analysis can be due to technical issues, such as lack of specific staining, low microscope quality, and insufficient expertise of the pathologists. It has been reported [6] that other urease-positive bacteria could be the cause for obtaining false-positive results. An option to overcome these limitations is PCR detection of *H. pylori* DNA in gastric mucosa, gastric juice, and stools. If specific primers to *H. pylori* are used when targeting more than one gene, PCR-based diagnostics may be considered as the gold standard test [7].

The rapid urease test (RUT) [8] detects indirectly the presence of *H. pylori* in a biopsy specimen obtained during gastroscopy. There are a variety of commercially available urease tests, including gel, paper and liquid-based assays with reaction times ranging from 24 h to several minutes. A strip test is convenient for use in endoscopy clinics and the RUT has the first-line indication to testing for diagnosis. The sensitivity of RUT is influenced by the bacterial density and the morphological forms (spiral or coccoid) of bacteria present in the biopsy. The presence of urease from other *H.* spp. in the stomach cannot be denied and false positives can occur.

The diagnostic methods mentioned above do not exhaust all standard methods used in practice. These are the most commonly used ones, and their advantages and disadvantages determine the effectiveness of the treatment. At the moment, it is considered that the combined application of some of the methods listed above can compensate for their shortcomings, as has been recommended by the diagnostic protocols.

### 1.2. Immunosensors

The limitations of traditional methods have promoted the development of innovative approaches for effective diagnosis of *H. pylori* infection—immunosensors based on the interaction between certain outer membrane proteins of the bacterial cell and certain cell receptors. Since the role of these interactions in bacterial colonization and pathogenesis is essential, they have been studied extensively; for reviews see [9,10] and the references cited therein.

Herein, we assess to what extent the reactions known currently can provide a basis for a reliable diagnosis of the infection.

The proteins that mediate the binding of *H. pylori* to the host cell receptors are BabA (blood-group-antigen-binding adhesin) binds Lewis b (Le^b^), SabA (sialic acid-binding adhesin, AlpA/B (adherence associated lipoprotein A and B), and OipA (outer inflammatory protein A). The enumerated proteins are the major virulence factors of *H. pylori* [11,12,13]. A major component of outer membrane proteins is an O-specific polysaccharide chain. The O-antigen of *H. pylori* outer proteins contains different human Lewis-like antigens, including Lewis (Le)x, Ley, Lea, and Le^b^, which are expressed in gastric epithelial cells [14].

SabA is a well characterized adhesin [15] with known multiple receptors expressed in gastric epithelia (as sialyl-Lewis X, sialyl-Lewis A, and Lex) or in saliva (as MUC7 and MUC5B). The application of these reactions for *H. pylori* detection is not reliable since the specificity is questionable.

The binding ability of AlpA/B to laminin was studied in [16]. The kinetics of the binding reaction was studied by means of surface plasmon resonance (SPR): laminin was immobilized on the biochip and bacterial suspensions applied to the SPR biochip by a flow cell. The binding reaction AlpA/B–laminin is non-specific and can hardly be used for sensing application.

BabA is the best characterized adhesin of *H. pylori* [17,18]. BabA adhesin binds to multiple receptors expressed in gastric epithelia and saliva. The BabA protein specifically binds by a covalent bond to Le^b^ and high specificity of the reaction was reported [17].

### 1.3. Biosensors

Although it has been shown that the best characterized receptor for *H. pylori* is the Lewisb (Le^b^) blood group antigen, the BabA–Le^b^ reaction was applied for the first time in our study [19] to detect bacterial infection. In this study we reported a direct test for detection of intact *H. pylori* bacteria, based on the BabA–Le^b^ binding reaction. An immunesensor intended for BabA adhesin detection was proposed in [20]. The sensors proposed [19,20] can be classified as biosensors—they combine a biorecognition component with a transducer. In [19] the biorecognition element was Le^b^ immobilized on SPR transducer; in [20] the transducer was an electrode imbedded in an electrochemical sensor; however, the recognition molecule was not strictly specified. The electrochemical sensor was applied to stool samples but unfortunately, the sensitivity achieved was not reported.

The significance of biosensors in the diagnosis of *H. pylori* infection is widely highlighted; for a review see [21,22] and the references cited therein. Detection of intact *H. pylori* cells was achieved by a graphene-based amperometric transducer [23], functionalized with bifunctional peptide, selective to binding *H. pylori*. The sensitivity registered was 100 CFU/mL. Other biosensors whose sensitivity was reported in terms of CFU/mL, actually register DNA, and from the average size of which the number of bacteria can be calculated [24]. Given the danger of food contamination with *H. pylori*, on-site control by biosensors is extremely important, as highlighted in [25].

In the present study we developed biosensors based on an SPR platform and a double resonance long period grating (DR LPG) platform. The biosensors were based on the BabA–Le^b^ binding reaction targeting diagnosing *H. pylori* infection by a direct observation of intact bacterial cells. The goal of this study was to demonstrate the feasibility and reliability of the BabA–Le^b^ binding reaction and to provide a promising platform for label-free detection of *H. pylori* infection with high selectivity and fast response.

## 2. Reagents and Materials

All the chemicals and reagents used were of analytical grade.

### 2.1. Bacterial Strain

*H. pylori* (DSM 21031) was obtained from the DSMZ-German Collection of Microorganisms and Cell Cultures, GmbH, Leibniz Institute, Inhoffenstrasse 7b, 38124 Braunschweig, Science Campus Braunschweig—Sued, Germany.

### 2.2. Cultivation Procedure

The procedure involved the following steps:

#### 2.2.1. Cultivation of Bacteria

Cultivation of the bacteria was performed in the course of 1–2 days at 37 °C in an anaerobic container equipped with a Campylobacter gas pack to create a suitable gas culture medium. A thin layer of brain heart infusion (Difco) liquid medium covered the surface of the agar medium—Columbia blood agar

#### 2.2.2. Preparation of Bacterial Suspensions

The preparation of the bacterial suspensions for treating the SPR biochip consisted of the following:preparation of a stock suspension with deionized water (density of 10^9^ CFU/mL);preparation of ten-fold dilutions of this suspension at concentrations ranging from 10^8^ to 10^2^ CFU/mL.

### 2.3. Blood Antigen

The blood antigen used was Lewis b tetrasaccharide, L7659 purchased from Sigma-Aldrich, 2909 Laclede Ave, St. Louis, MO 63103, United States.

## 3. Optical Platforms and Their Functionalization for Bacterial Sensing

### 3.1. SPR Platforms

The transducer converting the biochemical reaction into an optical signal of the surface plasmon (SP) wave was a gilded diffraction grating. The diffraction grating fulfills the matching between the photon momentum and the SP momentum at a certain angle of incidence of the excitation light and the wavelength. The gratings, having 90 nm high grooves at a distance of 1.20 µm from one another were supplied by DEMAX Ltd., Sofia, Bulgaria. At resonance conditions (excitation light at wavelengths in the 650–680 nm range, incidence angle of about 35 degrees), the SP wave was excited on the surface of the grating as shown schematically in Figure 1. The SPR biochip is a grating covered with a biorecognition layer of blood antigen Lewis b of certain thickness, as shown in Figure 1.

Details on our SPR measurement system based on spectrum readout can be found in [26]. The spectrometer accuracy measurement was 0.2 nm, while the overall accuracy of the resonance shift measurement of the SPR system was appraised at about 2 nm. The limit of detection (LOD) was evaluated by considering the accuracy of the spectrometer, as well as the accuracy of the goniometer for setting up the angle of light incidence.

### 3.2. Double Resonance Long Period Gratings (DR LPG)

The sensing platform used was a previously described [25,26] double resonance long period fiber grating (DR LPG). Such a grating is highly sensitive to the surrounding refractive index changes. A Lumonics Pulsemaster KrF excimer laser (238 nm), in combination with a chromium amplitude mask of a *Λ* = 135.6 μm period, was used to fabricate the gratings in a hydrogenated SM600 single mode fiber (cut-off wavelength *λ*_c_ ≈ 500 nm to 600 nm). The LPGs were made so that in air the minimum appeared around *λ*_TP_ = 774 nm, but when immersed in water the spectrum split into two minima at *λ*_L_ (left) and *λ*_R_ (right) separated by Δ*λ* which among others, depends on the surrounding refractive index (SRI) *n*, i.e., Δλ = Δλ(*n*). The dependence over a limited range can be represented as below:(1)Δλ(n)=Sn+A
where *S* is the sensitivity of the particular grating (nm/r.i.u.) and *A* is some constant.

The deposition of a nanolayer of Le^b^ blood antigen to functionalize the fiber surface slightly changed the surrounding refractive index and increased the separation Δ*λ*. When the DR LPG was immersed in water containing *H. pylori*, the bacteria were selectively attached to the functionalized fiber surface increasing the SRI by *δn*, which increases the separation Δ*λ* by *δλ*. Thus, the change of *δn* is indicative of the change in bacterial concentration. Figure 2a illustrates the spectral shifts δλ_L_ and δλ_R_ of the left and right resonance minima caused by the changes in the surrounding refractive index. The total wavelength shift is δλ = δλ_L_ + δλ_R_, the dependence of which on the SRI is presented in Figure 2b. The two slopes having different sensitivities (S_1_ = 792.82 nm/r.i.u and S_2_ = 438.92 nm/r.i.u) are observed for all LPGs used.

### 3.3. Functionalization of the Platforms

To ensure maximum specificity in bio-interactions, we immobilized the biorecognition molecules directly on the transducer without using a built-in matrix. For the purpose, we used the matrix-assisted pulsed laser evaporation (MAPLE) technology, which has proven its feasibility and reliability through the immobilization of hemoglobin, myoglobin, and antibodies [26,27,28]. Details regarding the MAPLE technique and the parameters of the immobilization procedure can be found in [28].

Our experimental experience with MAPLE technology established a relationship between the molecular weight of the ligand and the fluence, the parameter of the laser irradiation that determines the interaction with the substance subjected to evaporation. After the fluence was fixed by taking into consideration the Le^b^ molecular weight, the Le^b^ concentration of the solution was experimentally determined. This parameter determines the sensitivity of detection—the laser radiation inevitably defragments some part of the molecules, and the process is highly pronounced at higher concentrations. On the other hand, the small number of immobilized biorecognition molecules results in low sensitivity. Thus, the optimal concentration was established as a tradeoff between MAPLE technology requirements and detection sensitivity. The successful immobilization of Le^b^ was evaluated by the spectral displacement of the plasmon resonance after Le^b^ immobilization versus the resonance of a bare diffraction grating. The optimal concentration of the Le^b^ solution was determined by measuring the sensitivity of detection after the SPR biochip incubation with a bacterial suspension of a certain concentration. The optimal Le^b^ concentration was established at 0.2–0.4 mg/mL, at which the bioactivity of the molecules immobilized at given MAPLE parameters was preserved to a maximum extent.

The Le^b^ deposition was performed simultaneously on DR LPG and SPR transducers. The transducers were placed in a vacuum camera in groups of fifteen (twelve SPR + three DR LPG) and were subsequently functionalized by the MAPLE technique. Figure 3 below presents the MAPLE experimental setup for the simultaneous deposition of the blood antigen upon the DR LPG and the SPR DG sensing platforms. The pulse duration of the first harmonic of a YAG:Nd laser was 20 ns, the repetition rate 10 Hz. The laser fluency of our experiments varied from 100 mJ/cm^2^ to 300 mJ/cm^2^, with the laser light continuously scanning the target during the deposition process.

### 3.4. Characterization of the Deposited Film of Blood Antigen Molecules

Laser fluence plays a critical role, not only in conserving the chemical structure but also in film morphology. The aim was to deposit a dense film with good uniformity that would cover the gilded grating. This avoids a possible non-specific adsorption of the analyte on the metal surface and the non-uniform sensitivity over the biochip surface.

The structure of the deposited films was studied by TEM. Figure 4a,b shows images of the MAPLE-deposited film which has a good density but a low uniformity (Figure 4a), while cluster structure is also observed (Figure 4b).

The non-uniformities observed cause a dependence of the detection sensitivity on the position on the biochip. Certainly, they also determine different sensitivities for the different biochips which have different resonance spectral shifts after immobilization. The AFM micrograph (Figure 5) also reveals the cluster structure of the Le^b^ film.

### 3.5. Incubation of SPR Chips and Measurement Procedure

The biochips were incubated for 20 min at room temperature, followed by washing with deionized water (<2 µS/cm), after which the liquid phase was removed by centrifugation. All treatments were performed under the same conditions.

After the gilded diffraction gratings were functionalized, the plasmon resonances were measured at six different points on the biochip surface to evaluate the quality of the ligand layer. The spectral position of the resonances at each point was taken as a reference against which the shift resulting from the BabA–Le^b^ interaction was registered.

After incubation, the plasmon resonances were measured at the same six points on the biochip surface and the resonance wavelength shifts were estimated as differences from the reference resonances. Thus, the corresponding average displacement values and the absolute measurement errors were determined.

The procedure described above takes into account the inhomogeneity of the deposited layer for a given biochip but does not take into account the differences between the individual biochips. The sensitivity of the latter was not measured by a calibration procedure, but was estimated from the spectral shifts of the resonances following ligand immobilization.

### 3.6. Incubation of the LPG Structures and Measurement Procedure

Figure 6 presents the measurement set-up for detecting spectral changes and shifts δλ. A halogen lamp was used as a broadband source and the spectrometer (Avaspec) was sensitive in the 350 nm to 900 nm range. A small weight maintained a constant and reproducible strain along the LPG throughout all measurements.

## 4. Results and Discussion

### 4.1. Results of SPR Measurements

Figure 7a,b shows the resonance spectral displacement of the plasmon resonance resulting from the binding reaction between *H. pylori* and Le^b^ at different concentrations of the ligand.

The functional dependence on bacterial concentration was different at different ligand concentrations. This was well expressed at low bacterial concentrations; at the highest, no difference was observed given the accuracy of measurement. This is a reasonable result keeping in mind that the number of Le^b^ molecules bound to BabA was evaluated at 500 per bacterial cell [17].

It was experimentally established that higher Le^b^ concentration did not increase sensitivity of detection: the photothermal process inherent to MAPLE deposition was not limited only to the volatile solvent molecules—Le^b^ molecules were also involved and part of them was subsequently destroyed.

As Figure 7 indicates, a bacterial concentration lower than 2000 CFU/mL generates displacement below the limit of detection (LOD), accounting for measurement error (in this case of 0.6 nm). Therefore, the detection limit for SPR measurement was assessed at about 3000 CFU/mL.

### 4.2. Results of DR LPG Measurements

DR LPG transducers were elaborated simultaneously at one MAPLE deposition at 0.25 mg/mL Le^b^ concentration. We measured the spectral responses for three different bacterial concentrations 10^2^, 10^3^, and 10^4^ CFU/mL. Each concentration was measured with a different grating: DR LPG #1, DR LPG #2, and DR LPG #3. Since the gratings differed in characteristics, we measured the Δλ = Δλ(*n*) dependencies, from which the sensitivity S_i_ (nm/r.i.u.) and the constants A_i_ (i = 1, 2, 3) were obtained for *n* around the water index as follows: S_1_ = 470.68 nm/r.i.u., A_1_ = −626 nm, S_2_ = 481.25 nm/r.i.u., A_2_ = −640.3 nm, and S_3_ = 456.65 nm/r.i.u., A_3_ = −607.7 nm. Then the DR LPGs were functionalized following the procedure described above. Next, the spectra of a grating were determined in air and in water. After that the water was removed and the grating was submerged in a suspension with a particular bacterial concentration. Once in a suspension environment, the spectra were measured over the course of an hour. Figure 8a shows the spectra for DR LPG #1 at a concentration c_1_ = 10^2^ CFU/mL in air and water, at 1 min, 10 min, and 100 min after incubation, as well as in water after bacterial exposure. From these spectra we calculated the spectral shifts δλ at different moments t_i_ with respect to the spectral distribution in water and the results are presented in Figure 7b It can be seen from this figure that when the grating is in a suspension containing *H. pylori* an abrupt wavelength shift of about 2 nm occurs, followed by a slow rise over a span of one hour, which corresponds to the process of bacteria being trapped to the functionalized LPG surface. The responses of the other two LPGs for the concentrations c_2_ = 10^3^ CFU/mL (DR LPG #2) and c_3_ = 10^4^ CFU/mL (DR LPG #3) are similar to those shown in Figure 8a,b.

After one hour of exposure to a given concentration of bacteria the grating was washed, immersed in water, and the final spectrum measured. Since the bacteria were firmly bonded, we considered as meaningful the final wavelength shift in water. We thus compared the spectral shift in water in the presence and in the lack of bacteria. The corresponding resulting wavelength shifts for each one of the gratings were as follows: *δλ*_1_ = 2.39 nm, *δλ*_2_ = 2.46 nm, *δλ*_3_ = 2.37 nm. It follows from (1) that n = [Δλ − A]/S, so we found the refractive index for each concentration and plotted the dependence of the refractive index change vs. the bacterial concentration δn(c), as shown in Figure 9.

The LODs achieved in this study are close to the best reported to date: 88 CFU/mL achieved by aptamer-binding fluorescence methods [29] and 10 CFU/mL achieved by colorimetric detection [24].

## 5. Conclusions

We successfully investigated the applicability of the BabA–Le^b^ binding reaction for diagnosing *H. pylori* infection. With SPR detection, the limit of detection was estimated at 3000 CFU/mL. The sensitivity obtained in this way was higher than in the RUT sensitivity test, where a minimum of 10^4^ organisms per biopsy piece was required for a positive result [8]. Therefore, the SPR biosensor can be applied clinically for diagnosing *H. pylori* infection, given its potential to provide rapid and quantitative diagnosis. We would like to point out that the developed SPR sensor is the first of its kind for *H. pylori* detection, to the best of our knowledge.

The DR LPG sensing platforms provided much higher sensitivity—concentrations as low as 10^2^ CFU/mL were detected. The DR LPG measurements also showed an advantage in terms of accuracy. Essential for the DR LPG method is the requirement for an initial calibration procedure providing a reliable reference point. This explains the higher accuracy achieved, as compared to that of the SPR detection. The DR LPG detection is suitable for laboratory application since the measurement procedure is complicated.

When using SPR for detection, it is vital to have a stable reference before the biorecognition events start and this can only be provided by direct experimental measurement. The lack of such a reference (i.e., initial calibration) explains the large error and the high LOD in SPR-based measurement.

The higher accuracy of DR LPG sensing allowed the observation of the rise in the surrounding refractive index depending on the bacterial concentration. Real-time binding of *H. pylori* was observed. The kinetics of the process proves that the deposited Le^b^ layer has a preserved bioactivity.

## Figures and Tables

**Figure 1 sensors-24-02118-f001:**
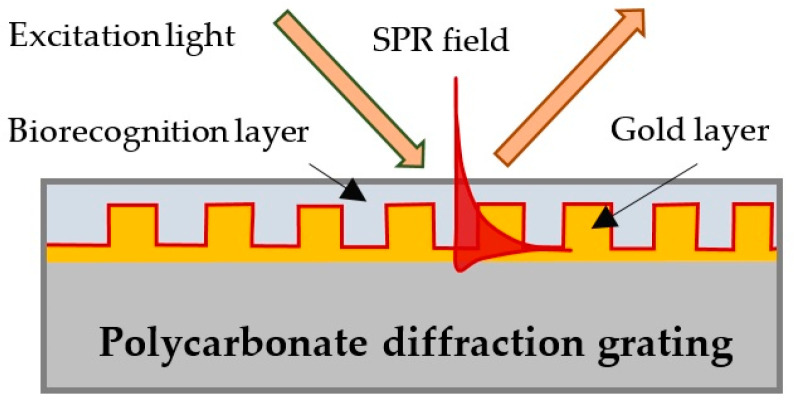
Surface plasmon wave excitation and a biochip—a gilded diffraction grating with a layer of immobilized Le^b^.

**Figure 2 sensors-24-02118-f002:**
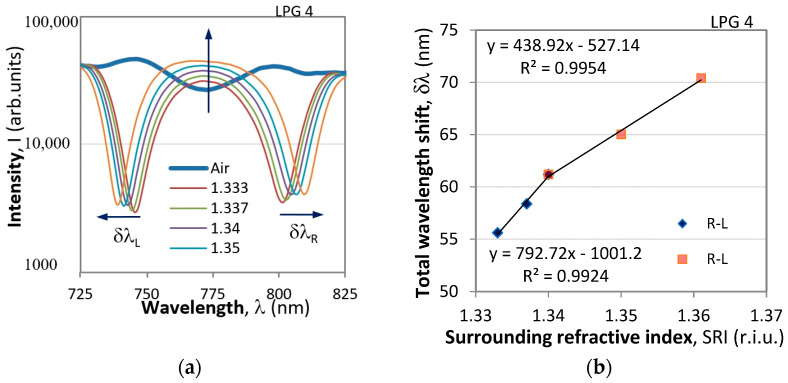
Spectral changes of a DR LPG caused by the surrounding refractive index (SRI) changes. (**a**) shift of the right and left minima, (**b**) total wavelength shifts and sensitivities.

**Figure 3 sensors-24-02118-f003:**
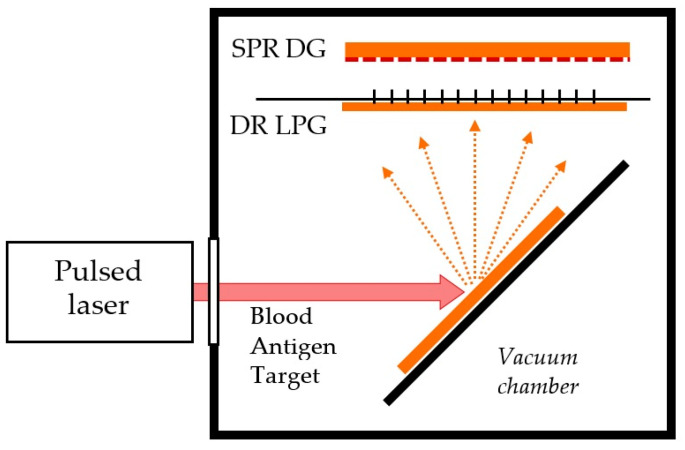
Schematic representation of the experimental setup for the MAPLE laser deposition of the blood antigen upon the DR LPG and SPR DR sensing platforms.

**Figure 4 sensors-24-02118-f004:**
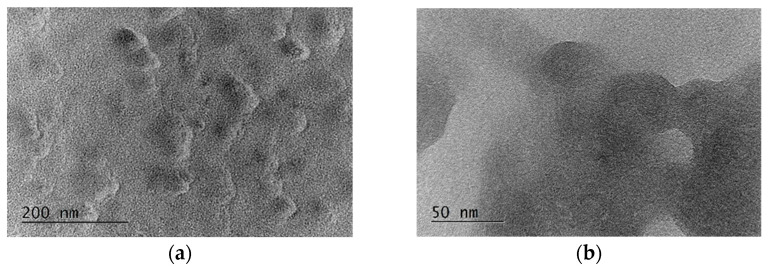
TEM images of: (**a**) a deposited layer; (**b**) Le^b^ cluster (a magnified part of Figure 4a).

**Figure 5 sensors-24-02118-f005:**
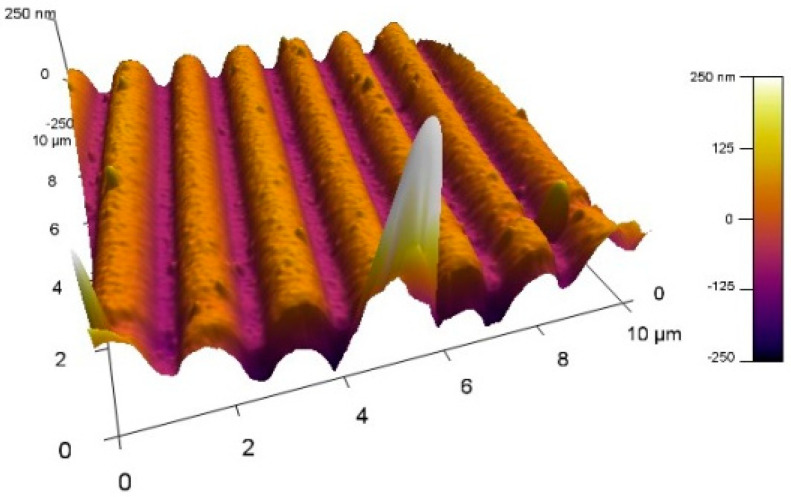
AFM image of the Le^b^ deposited film.

**Figure 6 sensors-24-02118-f006:**
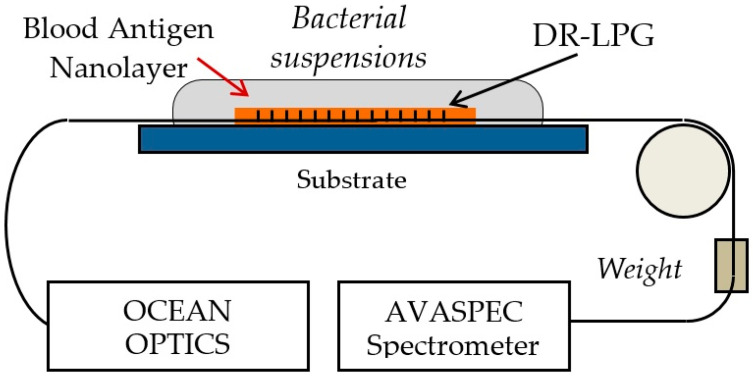
Set-up for measuring the spectral changes in a functionalized DR LPG during bacterial detection.

**Figure 7 sensors-24-02118-f007:**
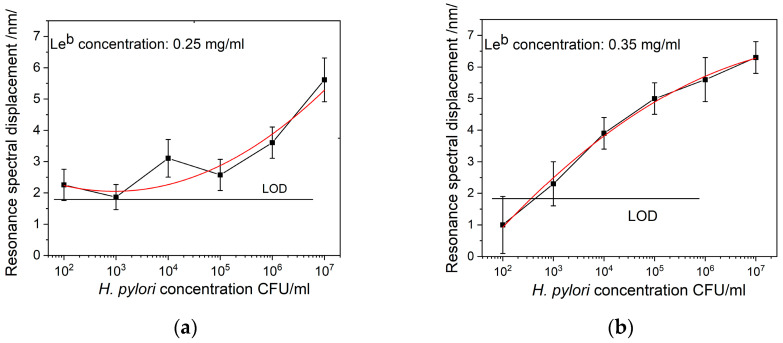
Displacement of resonance as a function of *H. pylori* concentrations for chips functionalized with (**a**) Le^b^ concentration 0.25 mg/mL, (**b**) Le^b^ concentration 0.35 mg/mL.

**Figure 8 sensors-24-02118-f008:**
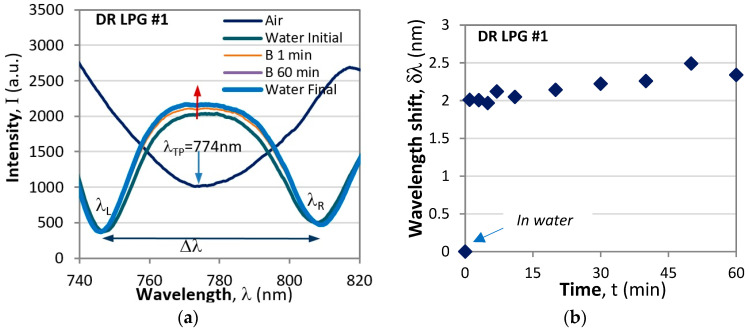
Spectral responses of DR LPG#1: (**a**) spectral distributions in air, water and in the presence of bacteria after 1 min (B1 min) and 60 min (B60 min) immersion; (**b**) wavelength shifts over a span of 60 min.

**Figure 9 sensors-24-02118-f009:**
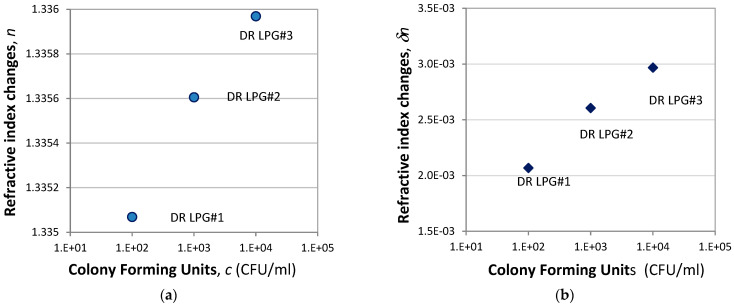
Dependence of the (**a**) refractive index change of the functionalized DR LPG on *H. pylori* concentration, (**b**) change of the refractive index with respect to water.

## Data Availability

Data are available after request to the authors.

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
