# Peer review of "SPR and Double Resonance LPG Biosensors for Helicobacter pylori BabA Antigen Detection"

_sensors, 2024, doi:10.3390/s24072118_

Round 1

Reviewer 1 Report

Comments and Suggestions for Authors

I checked the manuscript in some places, H. pylori, and in another place, Helicobacter pylori. Please unify all of them.

Please write a short introductory paragraph to the article before the subheadings. Revise the information under subheadings to make it a little shorter and increase the general reader's interest in the article.

Please provide some more details on material characterization (TEM and AFM). Do the properties of the material contribute to sensor selectivity? Using the ellipsometry technique for characterization will provide more detailed information about the material. I recommend trying it if possible.

Please rewrite the conclusion section. Instead of giving it in bullet points, prepare it as a whole paragraph. Instead of representing the experiment results, give information about the expectations and possible usage potential in light of the results obtained.

The similarity rate of the article is seen as 25%. I think this rate is very high in research articles, provided that it is the decision of the journal editor. When the article is examined, it is seen that many parts show overall similarity.

Reviewer 2 Report

Comments and Suggestions for Authors

The manuscript provided SPR and double resonance LPG biosensors. The binding reaction of BabA-Leb were employed to capture the H. pylori. The SPR biosensor exhibited an acceptable sensitivity of 3000 CFU/ml. The double resonance LPG biosensors provide a higher sensitivity of 100 CFU/ml. The manuscript was suggested to be accepted after a major revision. The issues were listed as the following:

1.     The innovation of the biosensor was suggested to be highlighted in the abstract. Moreover, the meaning of the biosensor was also added in the ending of the abstract.

2.     There are lots of biosensors for H. pylori detection, including fiber optic-based, fluorescent, colorimetric, and electrochemical biosensors. Recently, a mini review (J. Agric. Food Chem. 2023, 71, 1098210988) summarized the significant process of the fiber optic-based biosensor for bacteria detection. The recently work was suggested to be cited in the manuscript.

3.     Two dips were exhibited a significant wavelength shift with the surrounding refractive index in Figure 2. The linear curves of the two dips were suggested to be given in the Figure 2. The slopes of the linear curves were also calculated and discussed in the manuscript.

4.     The element analysis was suggested to be given in Figure 4.

5.     The anti-interference experiment was performed in the manuscript.

Reviewer 3 Report

Comments and Suggestions for Authors

The authors present a manuscript which provides an original approach for the development of theSPR-based sensor for Helicobacter pylori BabA Antigen Detection. The design of the experiment is explained in detail and the reported results seem to be correct. The authors developed developed Surface Plasmon Resonance (SPR) and Double Resonance Long Period Grating (DR LPG) biosensors and performed its functionalization with blood antigen molecules by MAPLE technique. The goal of the study was to monitor the BabA – LEb binding reaction to realize the label-free detection of H. Pylori and this goal was achieved. The two different platforms have provided different sensitivities and LODs: 3000 CFU/ml and 100 CFU/ml for SPR and DR LPG respectively. Despite all advantages, the quality of the manuscript can be significantly improved.

1) First of all, the choice of sensing platforms should be clarified. Thus, the authors call the diffraction grating lattice resonance platform a SPR, while traditionally the SPR is a plasmon-polariton excited over the thin metal/dielectric in ATR geometry. The surface lattice resonances are known to provide excellent phase sensitivity due to topological darkness effect [1] while its spectral sensitivity (nm/RIU) is limited to its period. For example, for the 1um period grating the spectral sensitivity will be on the order of 1000nm/RIU, while commercially used SPR sensors provide much higher spectral sensitivity (on the order of 10^4 nm/RIU). What was the reason of choosing the diffraction grating lattice resonance platform instead of pure SPR sensor?

2) Then, the obtained LODs of proposed biosensors should be compared with ones obtained with other sensing techniques.

Despite this issues the overall quality of the paper is good and it can be considered for publication in Sensors after minor revision.

Reviewer 4 Report

Comments and Suggestions for Authors

Interesting work. Well done. I did not observe real-time binding curves for the interaction of H pylori. In this way you can show mass transport limitation of the bacteria and/or inhomogeneous binding processes. This will improve the paper. I found a misspelled MALPE instead of MAPLE. 

Round 2

Reviewer 1 Report

Comments and Suggestions for Authors

The author edited the article in line with the suggestions. The article is acceptable in its current form.

Reviewer 2 Report

Comments and Suggestions for Authors

The manuscript has been greatly improved comparing with the original version. Thus, the revised version could be accepted by the journal of Sensors.

Reviewer 3 Report

Comments and Suggestions for Authors

The authors have addressed my concerns sufficiently in the revised manuscript and have improved the quality of the manuscript